# SciPG: A New Benchmark and Approach for Layout-aware Scientific Poster Generation

## Abstract

Scientific posters are an effective and expressive medium for conveying the core ideas of academic papers, facilitating the communication of research techniques. However, creating high-quality scientific posters is a complex and time-consuming task that requires advanced skills to summarize key concepts and arrange them logically and visually appealingly. Previous studies have primarily focused on either content extraction or the layout and composition of posters, often relying on small-scale datasets. The scarcity of publicly available datasets has further limited advancements in this field. In this paper, we introduce a new task called layout-aware scientific poster generation (LayoutSciPG), which aims to generate *flexible* posters from scientific papers through integrated automatic content extraction and layout design. To achieve this, we first build a new dataset, namely SciPG, containing over 10,000 pairs of scientific papers and their corresponding posters. We then propose a multimodal extractor-generator framework, which employs a multimodal extractor to retrieve key text and image elements from the papers and designs an interactive generator with an adaptive memory mechanism to seamlessly paraphrase the extracted content and generate a structured layout. This approach effectively tackles challenges related to GPU memory consumption and long-term dependencies when handling the lengthy inputs (scientific papers) and outputs (posters). Finally, both qualitative and quantitative evaluations demonstrate the effectiveness of our approach while highlighting remaining challenges.

## 1 Introduction

Recent years have witnessed a significant increase in the number of scientific papers published in various academic conferences and journals. For example, AAAI, a prominent international conference on artificial intelligence, received over 2,300 papers in 2024 alone. It is time-consuming for researchers to digest all these papers. Scientific posters offer an effective and expressive way to present the core ideas and findings from original papers, enabling researchers to quickly grasp the overall content. However, creating a high-quality scientific poster from scratch that is both informative and aesthetically pleasing is a challenging task. Poster design is a complex and time-consuming task that requires both a deep understanding of the paper's content and experience in design. Consequently, the need for automatic generation of readable, informative, and visually appealing posters has become increasingly important.

Scientific poster generation involves multimodal understanding and reasoning, as both the paper document and the poster contain tightly integrated text and image elements [1]. Previous approaches for automatic poster generation have primarily focused on either the layout and composition of posters (Paramita & Khodra, 2016; Qiang et al., 2016; 2019) or content extraction (Xu & Wan, 2021; 2022). These methods have not performed well because those emphasizing layout and composition often neglect the importance of robust content extraction, relying on simple summarization models such as TextRank (Mihalcea & Tarau, 2004). Conversely, approaches that focus on content extraction typically use LaTeX templates for poster generation, which lack diversity and flexibility

---

[1]The image element includes figures, charts, and tables; henceforth, we will refer to them collectively as images.

in layout design. These naturally bring us to a question: can machines learn to automatically generate *diverse* and *flexible* posters from a large quantity of example pairs of papers and posters created by human experts?

Table 1: A summary of main scientific poster generation datasets. Compared to ours, these datasets are relatively small. Moreover, they are typically designed for either content extraction or layout generation, rather than encompassing both aspects.

| Datasets | #(Paper-Poster Pairs) | Tasks | |
| | | Content Extraction | Layout Generation |
| --- | --- | --- | --- |
| NCE ((Xu & Wan, 2021)) | 60 | ✓ | ✗ |
| PGM ((Qiang et al., 2016)) | 25 | ✗ | ✓ |
| NJU-Fudan ((Qiang et al., 2019)) | 85 | ✗ | ✓ |
| SciPG | 11,302 | ✓ | ✓ |

To automatically generate a diverse and flexible scientific poster that accurately represents the original paper, three key challenges need to be addressed: (1) *Multimodal Extraction*: Important text and image elements must be exactly extracted from the original paper. (2) *Multimodal Generation*: The extracted textual elements typically cannot be directly placed onto the poster. They need to be paraphrased into a concise form suitable for the poster. Additionally, the size and placement of both the extracted image elements and the paraphrased text elements must be carefully considered. (3) *Large-Scale Training Data*: As shown in Table 1, existing data-driven approaches rely on small-scale datasets. The lack of publicly available large-scale datasets has hindered further research in this area.

To address the aforementioned challenges, we first collect a new dataset of paper-poster pairs from public conference web pages, explicitly aligning elements of each paper with its corresponding poster. We then propose a multimodal extractor-generator framework for LayoutSciPG, which involving: (1) Multimodal Extraction: Using a multimodal extractor to retrieve text and image elements from the paper. (2) Interactive Generation: Implementing an interactive generator with an adaptive memory mechanism to jointly paraphrase the extracted elements and generate the corresponding layout positions. This interactive generation and adaptive memory mechanism address the challenges of GPU memory cost and long-term dependencies in handling the lengthy inputs (papers) and outputs (posters).

The contributions of this paper are as follows:

- We create and will release a new dataset [2], namely SciPG, for the task of scientific poster generation for research purposes.

- We develop a multimodal extractor-generator framework for LayoutSciPG. This includes a multimodal extractor for joint text and image extraction, and an interactive generator that unifies the paraphrasing of extracted elements and layout generation.

- Both automatic and human evaluation results demonstrate the effectiveness of our approach, while also revealing some remaining challenges.

## 2 RELATED WORK

### 2.1 TEXT SUMMARIZATION

Text summarization generally falls into two categories: *extractive summarization* (Cheng & Lapata, 2016; Yao et al., 2018; Nallapati et al., 2017) and *abstractive summarization* (Nallapati et al., 2016; Yao et al., 2020; See et al., 2017). Extractive summarization focuses on identifying the most salient parts of the input document and using them directly as the output summary. For example, Cheng & Lapata (2016) employed a neural attention model to select sentences or words from the input document as the summary. Similarly, Nallapati et al. (2017) developed a recurrent neural network (RNN) for extractive summarization. While abstractive summarization involves paraphrasing or rewriting the important parts of the input document into a concise summary. For instance, Lead3 (Nallapati et al., 2016) used an attentional encoder-decoder RNN for abstractive text summarization.

---

[2]The source code and data will be made available upon acceptance of this work.

See et al. (2017) designed a hybrid pointer-generator network that can copy words from the source text via pointing, enabling more robust summary generation. However, these approaches are tailored for pure text summarization and do not account for the graphical elements in the original document.

Our LayoutSciPG combines both abstractive and extractive summarization, as it requires extracting key elements from a document and rewriting it into a concise form. A closely related task is scientific document summarization (Jaidka et al., 2016; Parveen et al., 2016), but existing work in that area has primarily focused on producing pure text summaries. In contrast, our focus is on generating multimodal scientific posters.

## 2.2 MULTIMODAL SUMMARIZATION

Our LayoutSciPG closely aligns with multimodal summarization, which focuses on extracting the most important information from various modalities to create summaries. For example, Zhu et al. (2018) introduced the first multimodal summarization (MSMO) model and compiled a dataset that includes both text and image modalities. Li et al. (2020b) extended this approach to video-based news articles, employing conditional self-attention for text and video fusion. More recently, He et al. (2023) proposed a unified transformer-based model that effectively aligns and attends to multimodal inputs. While LayoutSciPG similarly involves summarizing multimodal documents, it also requires structuring the multimodal summary within a specific layout, incorporating layout prediction as an essential component.

## 2.3 AUTOMATIC POSTER GENERATION

Existing methods for poster generation have primarily concentrated on either the scientific poster composition (Paramita & Khodra, 2016; Qiang et al., 2016; 2019) or content extraction (Xu & Wan, 2021; 2022). Approaches that emphasize layout composition often overlook the importance of robust content extraction, relying on basic summarization models like TextRank (Mihalcea & Tarau, 2004). In addition, they have typically employed simple probabilistic graphical models to infer panel attributes, but these approaches require human annotation of poster panels. On the other hand, content-focused methods typically use predefined LaTeX templates for poster generation, which limits both the flexibility and aesthetic appeal of the resulting scientific posters. Additionally, current datasets in this domain are relatively small, with fewer than 300 paper-poster pairs each. To address this, we have constructed a new dataset comprising 11,302 pairs of high-quality scientific documents and posters.

## 3 DATASET

We collect pairs of scientific paper documents and the corresponding posters from the recent academic conference proceedings, including CVPR, ICML, NeurIPS and ICLR. These academic proceedings mainly focus on research communities of computer vision and machine learning. Table 2 reports the descriptive statistics of our dataset.

Table 2: Statistics of datasets. We report the average number of sentences, words and figures in a document or a poster.

| Conference | Year | Document - Poster Train / Val / Test | Documents | | | Posters | | |
|---|---|---|---|---|---|---|---|---|
| | | | #Sentences | #Words | #Figures | #Sentences | #Words | #Figures |
| CVPR | 2023 | 1,851 / 231 / 231 | 489.28 | 7827.48 | 10.82 | 38.29 | 345.80 | 8.14 |
| ICML | 2022 / 2023 | 1,910 / 239 / 239 | 799.5 | 13,397.18 | 15.13 | 55.03 | 572.79 | 5.94 |
| NeurIPS | 2022 / 2023 | 3,625 / 453 /453 | 634.59 | 10,712.07 | 11.43 | 51.9 | 544.55 | 6.52 |
| ICLR | 2023 / 2024 | 1,653 / 207 / 207 | 800.59 | 12,436.17 | 19.86 | 49.69 | 500.96 | 7.08 |
| Total | - | 9,039 / 1,129 /1,134 | 670.05 | 11,004.36 | 13.63 | 49.37 | 501.85 | 6.83 |

For the collected pairs, we randomly split them by 8:1:1, resulting in 9,039, 1,130, and 1,130 pairs allocated to the training, validation, and test sets, respectively. Meanwhile, we automatically extract text and image elements from documents and posters and perform matching to create document-to-poster alignment. The details for data processing and element alignment are represented in Appendix 6.

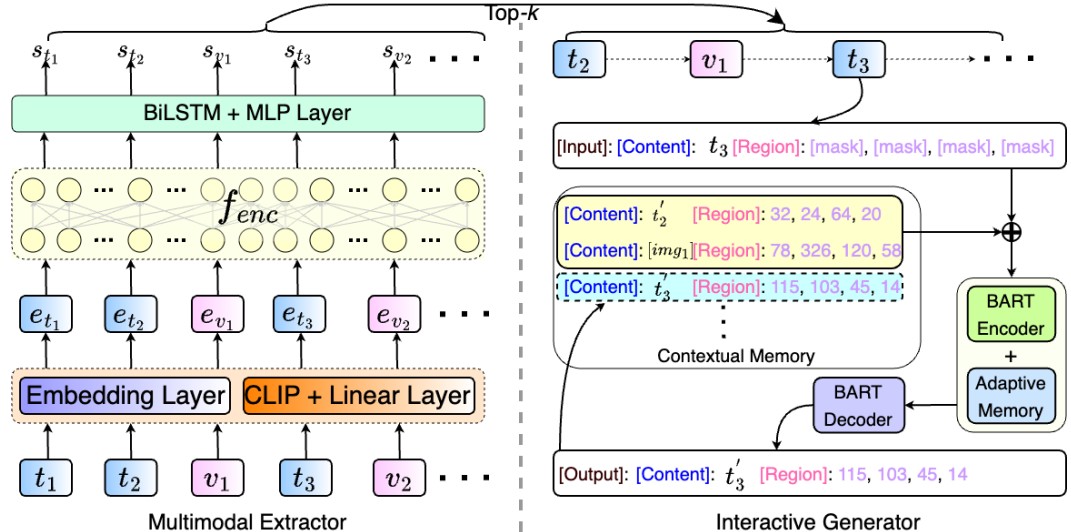

Figure 1: Overview of our proposed framework.

## 4 METHODOLOGY

### 4.1 OVERVIEW

LayoutSciPG aims to generate a poster from a multimodal document containing both text and images. We decompose this task into several subtasks: first, summarizing the document by extracting important sentences and images; next, paraphrasing the extracted sentences into a concise format suitable for poster presentation; and finally, placing the paraphrased sentences and images in appropriate locations on the poster. To achieve this, we propose a multimodal extractor-generator framework, which performs multimodal content extraction, paraphrasing, and layout generation. Figure 1 provides an overview of our approach, which includes the following modules:

- A **Multimodal Document Extractor (MDE)** encodes both sentences and images from the paper document and selects which of them should be extracted;

- An **Interactive Generator (IG)** fulfills both paraphrasing the extracted sentences and layout generation of text and image elements.

Given a scientific paper document $D = \{T, V\}$, where $T = \{t_1, t_2, ..., t_n\}$ is a sequence of $n$ sentences and $V = \{v_1, v_2, ..., v_m\}$ is a collection of $m$ images. MDE first extracts the important multimodal elements $X = \{X_t, X_v\}$ from $D$, and IG paraphrases the selected sentences $X_t$ into $Y_t$ and predicts the layout information $L = \{L_t, L_v\}$ of $\{Y_t, X_v\}$, where $X_t$ and $Y_t$ denote the directly extracted and paraphrased sentences, respectively. $X_v$ is a subset of images extracted from $D$. $L_t = \{x_0^t, y_0^t, w_0^t, h_0^t\}$ and $L_v = \{x_0^v, y_0^v, w_0^v, h_0^v\}$ are the bounding box (bbox) coordinates of the $Y_t$ and $X_v$, where the first two values indicate the top left corner location, and the last two indicate the width and height.

### 4.2 MULTIMODAL DOCUMENT EXTRACTOR

To capture both the textual and visual information from document $D$, we extend a text encoder, specifically RoBERTa (Liu et al., 2019), into a multimodal encoder. This multimodal encoder takes the concatenation of text embeddings ($e_t$) and visual embeddings ($e_v$) as input, outputting contextualized joint representations. Specifically, given a document $D$ containing $n$ sentences $\{t_1, t_2, ..., t_n\}$ and $m$ images $\{v_1, v_2, ..., v_m\}$, we use RoBERTa to encode each sentence $t_i$. For the images, we employ a pre-trained CLIP (Radford et al., 2021) vision encoder to extract visual features, which are then projected via a linear layer to match the hidden dimension of the text input. Additionally, we represent the images with a series of special index tokens: "[img_1]" denotes the first image, "[img_2]" the second, and so on. This formulation allows us to utilize the standard Transformer ar-

chitecture with minimal modifications while also facilitating the subsequent paraphrasing of visual elements. The encoding process for each modality is formulated as follows:

$$
\begin{aligned}
e_t &= \text{EmbeddingLayer}([t_1, ..., t_n]), \\
e_v^i &= \text{EmbeddingLayer}([\text{img\_1}], ..., [\text{img\_m}]), \\
e_v^j &= [\text{CLIP}(v_1), ..., \text{CLIP}(v_m)]W_v, \\
e &= [e_t; e_v] = [e_t, e_v^i + e_v^j], \\
h &= [h_t; h_v] = f_{enc}(e),
\end{aligned}
\tag{1}
$$

where EmbeddingLayer is the embedding layer of RoBERTa, $f_{enc}$ denotes RoBERTa based encoder function, and $W_v$ denotes the feature projection, which are learnable parameters.

In our encoder, we represent individual sentences by inserting additional "[CLS]" and "[SEP]" tokens at the beginning and end of each sentence, respectively. The "[CLS]" token is used to aggregate the features of the sentence that follows it, i.e., $\hat{h}_t = h_{t[CLS]}$. For visual features, we directly use $h_v$ as the representation for each image. Next, we apply a bidirectional long short-term memory network (BiLSTM) (Zhang et al., 2015) to capture the contextualized representations of both sentences and images. Finally, a fully connected layer (MLP) is used on top of the BiLSTM to predict an extractive score for each sentence and image:

$$
\begin{aligned}
[h_t^{'}; h_v^{'}] &= \text{BiLSTM}([\hat{h}_t; h_v]), \\
f_{cls}(h^{'}) &= W_{tv}[h_t^{'}; h_v^{'}] + b_{tv},
\end{aligned}
\tag{2}
$$

where $W_{tv}$ and $b_{tv}$ are learnable parameters. By adopting such a hierarchical framework for the semantic representations. RoBERTa at the lower level learns sentence-level semantic representations, while a BiLSTM at the higher level captures the contextual semantic representations for the entire document. Finally, based on the extractive score $s$ for each sentence and image, we use the standard binary cross-entropy loss as the objective function for the extractive references:

$$
\begin{aligned}
p_{tv}(\tilde{h}) &= \frac{\exp(f_{cls}(\tilde{h}))}{\sum_{\tilde{h} \in [h_t^{'}; h_v^{'}]} \exp(f_{cls}(\tilde{h}))}, \\
\mathcal{L}_{ext} &= - \sum_{\tilde{h} \in [h_t^{'}; h_v^{'}]} \log p_{tv}(\tilde{h})
\end{aligned}
\tag{3}
$$

## 4.3 INTERACTIVE GENERATOR

We propose an interactive generator that unifies multimodal generation (i.e., text, images, and layout) using BART (Lewis et al., 2019) in an interactive fashion. Given an input list from the extractor, at each step, we automatically feed one element of the input into the generator, which then returns its predicted result. This process is repeated iteratively until the entire input list has been processed. Instead of relying on a dedicated module for image representation, we adopt a similar approach as in the extractor by utilizing an index token, such as "[img_i]", to indicate that the $i$-th image is the target. This allows all target outputs to be transformed into a textual format, while the visual input are represented as the image feature, which are concatenated with textual token embeddings as input for the generator. Additionally, we encode the bounding boxes for each target sentence and image in the format "$x_0, y_0, w, h$", appended at the end of each sequence as the final target output, where $(x_0, y_0)$ represents the top-left corner coordinates and $(w, h)$ denotes the width and height. An example of the input-to-output format is provided on the right side of Figure 1. Note that we use four "[mask]" tokens to indicate the layout coordinates in the input, ensuring alignment with the pretraining task. To address the challenge of long target generation, we draw inspiration from the memory mechanism in the Recurrent Memory Transformer (RMT) (Bulatov et al., 2022) and introduce an adaptive memory module. This memory module enhances the model's ability to handle long-term dependencies in extended generation tasks, while the interactive generation approach mitigates GPU memory limitations.

### 4.3.1 GENERATOR FORMULATION.

Given the extracted elements $X = \{X_t, X_v\}$, For each element (text sentence or image) $x \in X$ , the generator predicts the generation probability $P_\theta(y_t, l_t | x, y_{<t}, l_{<t})$ based on the current input

element $x_t$ and previous contextual history $(y_{<t}, l_{<t})$. For the whole extracted elements $X$, the generation probability can be computed as: $P_\theta(Y, L|X)$

$$P_\theta(Y, L|X) = \prod_{x_t \in X} P_\theta(y_t, l_t|x_t, y_{<t}, l_{<t}) \tag{4}$$

where $y \in Y$ is the paraphrasing content of $x \in X$ and $l \in L$ is the corresponding bounding boxes of each extracted element $y$. Finally, our training objective employs negative log-likelihood (NLL) loss, combined with KL-Divergence between the predicted probability $P_\theta$ and the output one-hot distribution $P_\theta'$, incorporating Label Smoothing (Szegedy et al., 2016) to prevent the model from becoming overconfident.

$$\mathcal{L}_{gen} = -\log P_\theta(Y, L|X) + \beta * D_{KL}(P_\theta||P_\theta') \tag{5}$$

where $\beta$ is a weight parameter to balance the NLL loss and KL-D loss. During the interactive generation process, given a extracted element $x_t$, each step produces a new pair $(y_t, l_t)$, causing the contextual history to grow increasingly long. This accumulation leads to high GPU memory consumption and challenges in handling long-term dependencies. To address these issues, we introduce an adaptive memory mechanism.

### 4.3.2 ADAPTIVE MEMORY MECHANISM.

Inspired by the memory module in the RMT (Bulatov et al., 2022), we adapt it in the encoding process of our interactive generator. This adaptation augments the interactive generation process with adaptive memory, composed of $k$ real-valued trainable vectors. Specifically, at each step, the contextual history $(y_{<t}, l_{<t})$ is divided into $z$ segments, and memory vectors are prepended to the first segment embeddings and processed alongside the segment tokens. At the time step $\tau$ ($\tau \leq z$) and segment $H_\tau^0$, the recurrent step is performed as follows:

$$[\hat{H}_\tau^{mem}; H_\tau^N] = f_{enc}^g([H_\tau^{mem}; H_\tau^0]), \tag{6}$$

where $f_{enc}^g$ denotes the encoding process of our generator and $N$ is a number of encoder layers.

After the forward pass, $\hat{H}_\tau^{mem}$ contains updated memory tokens for the segment $\tau$. Segments of the input sequence are processed sequentially. To enable the recurrent connection, we introduce a MultiHeadAttention mechanism (Vaswani et al., 2017) to update the memory between the output memory tokens of the current segment and the input memory tokens of the next segment, enabling an adaptive attention over previous memories:

$$\begin{aligned} H_{\tau+1}^{mem} &= \text{MultiHeadAttention}(H_{\tau+1}^{mem}, \hat{H}_\tau^{mem}), \\ [\hat{H}_{\tau+1}^{mem}; H_{\tau+1}^N] &= f_{enc}^g([H_{\tau+1}^{mem}; H_{\tau+1}^0]), \end{aligned} \tag{7}$$

### 4.3.3 PRE-TRAINING OBJECTIVES.

As shown in Figure 1, flattening layout information into a text sequence often leads to the loss of spatial context, making it challenging for the generative model to understand the relationship between generated content and its spatial positioning. To improve the model's spatial awareness, we introduce several innovative self-supervised learning objectives for the extracted sequences in the posters. These sequences consist of OCR text blocks or visual tokens along with their corresponding bounding boxes. In the rest of this subsection, we introduce three sentinel tokens "[content]", "[region]" and "[mask]" and demonstrate their use with the following input text example:

"[content]: simple self-supervised learning of periodic targets [region]: 725, 52, 236, 50"

**(1) Joint Text-Layout Reconstruction** requires the generative model to simultaneously reconstruct missing text and predict the layout of entire text blocks. Specifically, we mask a portion of text tokens and all layout coordinates, tasking the model with reconstructing both the text and their corresponding bounding boxes (i.e., layout tokens). For example, assuming the words "simple" and "learning of" are masked, the input and target sequences would appear in the following table. Unlike the 15% masking ratio used in Masked Language Modeling (MLM) (Devlin et al., 2018), joint text-layout reconstruction employs a higher masking ratio of 50% for text. This is because using a smaller ratio would make the task too simple, whereas a larger ratio increases the difficulty and encourages more effective learning.

| |
|---|
| **Input Sequence:** |
| "[content]: [mask] self-supervised [mask] periodic targets [region]: [mask], [mask], [mask], [mask]" |
| **Output Sequence:** |
| "[content]: simple self-supervised learning of periodic targets [box]: 725, 52, 236, 50" |

**(2) Layout Modeling** requires the generative model to predict the spatial positions of a given text block based on the surrounding context. For instance, the model is tasked with predicting the coordinates of the top-left corner or the width and height of the text block. The input and target sequences are structured as follows:

| |
|---|
| **Input Sequence 1:** |
| "[content]: simple self-supervised learning of periodic targets [region]: [mask], [mask], 236, 50" |
| **Output Sequence 1:** |
| "[content]: simple self-supervised learning of periodic targets [region]: 725, 52, 236, 50" |
| **Input Sequence 2:** |
| "[content]: simple self-supervised learning of periodic targets [region]: 725, 52, [mask], [mask]" |
| **Output Sequence 2:** |
| "[content]: simple self-supervised learning of periodic targets [region]: 725, 52, 236, 50" |

**(3) Text Construction** involves generating a text sequence for a specified location on the poster. For example, if all text content is masked, the input and target sequences are as follows:

| |
|---|
| **Input Sequence:** |
| "[content]: [mask] [region]: 725, 52, 236, 50" |
| **Output Sequence:** |
| "[content]: simple self-supervised learning of periodic targets [region]: 725, 52, 236, 50" |

### 4.3.4 DATA EXTENSION.

During interactive generation, the input order of extracted elements directly impacts the generator's performance. Therefore, we sort the elements according to their order in the original document $D$, as the content in the poster is typically organized sequentially based on the document. Additionally, we shuffle the extracted elements to create supplementary extended data.

## 5 EXPERIMENTS

### 5.1 EVALUATION METRICS

LayoutSciPG is a multimodal extraction and generation task that involves producing textual, pictorial, and layout outputs. To assess the quality of each output type, we define distinct metrics for evaluation, as outlined below.

**ROUGE:** For the textual output, we report the F1 ROUGE score via ROUGE1.5.5.pl (Lin, 2004) which calculates the overlap lexical units between generated and ground-truth sentences. This metric includes *ROUGE-1*, *ROUGE-2* and *ROUGE-L*.

**ImgP** and **ImgR:** For the extracted images, we use *ImgP*) and *ImgR*) to evaluate the images selected by our method. ImgR represents the recall of extracted image elements, calculated as $\frac{\text{The number of correct images}}{\text{The total number of ground truth images}}$. ImgP represents the precision of extracted image elements, calculated as $\frac{\text{The number of correct images}}{\text{The total number of extracted images}}$.

To quantitatively evaluate layout performance, we introduce the following metrics:

**Overlap** and **Coverage:** Overlap refers to the intersection over union (IoU) of various layout elements. Generally, these elements do not overlap significantly, so the overlap score tends to be low. Coverage measures the percentage of the canvas occupied by the layout elements. These metrics help assess the spatial arrangement and utilization of space in the generated layouts.

**Validity**, **Alignment**, **FD** and **DreamSim**: Validity, annotated as Val, is the ratio of valid elements greater than 0.1% of the canvas. Alignment, annotated as Ali, is the extent of spatial non-alignment between elements. FD denotes the Frechet distance. DreamSim (Fu et al., 2023) is a perceptual metric that assesses the poster images holistically.

## 5.2 BASELINES

For the multimodal content extraction, we employ there baselines: *NeuralExt* (Xu & Wan, 2021) , *MSMO* (Zhu et al., 2018) and *AdaD2P* (Fu et al., 2022). NeuralExt is a neural extractive model designed to extract text, figures, and tables from a paper. MSMO is a multimodal attention model that jointly generates text and selects the most relevant image from multimodal input. We adapt MSMO for the content extraction task. AdaD2P is originally designed for document-to-slide generation. We adapt its the extraction module for our multimodal content extraction. For the multimodal generation, there are no established baselines to compare. We still adapt the content paraphrasing and layout prediction modules of AdaD2P for our task.

## 5.3 IMPLEMENTATION DETAILS

For the multimodal extractor, we initialize the encoder using the RoBERTa-base model [3], which consists of 12 layers, a hidden size of 768 dimensions, and 12 attention heads. The Bi-LSTM is configured with 768 hidden units. For the interactive generator, we initialize its parameters with the BART-large model [4]. The memory size is set to 50, and the KL-Divergence weight is set to 0.5. Our framework is trained using the ADAM (Kingma & Ba, 2015) optimizer with a learning rate of 3e-4. In the training phase, every 1000 iterations, we evaluate the model's performance on the validation dataset using the current parameters. After completing the training process, we load the model with the optimal parameters, as determined on the validation dataset, and test it on the test dataset. All experiments are carried out with Pytorch framework and fourNVIDIA A100-PCIE-40GB GPUs.

Table 3: The evaluation for multimodal element extraction. The best result is in boldface.

| Methods | Text | | | Image | |
|---|---|---|---|---|---|
| | ROUGE-1 | ROUGE-2 | ROUGE-L | ImgP | ImgR |
| NeuralExt | 36.55 | 12.67 | 14.43 | 31.68 | 24.91 |
| MSMO | 32.45 | 10.43 | 12.51 | 36.64 | 32.56 |
| AdaD2P | 38.28 | 13.04 | 15.72 | 38.24 | 33.76 |
| MDE | **40.68** | **14.76** | **17.54** | **44.43** | **40.57** |
| MDE w/o LSTM | 36.73 | 11.78 | 14.59 | 40.26 | 36.55 |

Table 4: The evaluation for multimodal generation. For the layout, the values in parentheses represent the ground-truth posters. All values in the table are expressed as percentages, with the best results highlighted in bold.

| Methods | Text | | | Layout | | |
|---|---|---|---|---|---|---|
| | ROUGE-1 | ROUGE-2 | ROUGE-L | Overlap | Coverage | |
| AdaD2P | 39.84 | 13.75 | 16.68 | 47.44 (5.11) | 12.38 (53.42) | |
| IG | **41.05** | **15.19** | **18.84** | **25.08** (5.11) | **37.43** (53.42) | |

Table 5: The experimental results for the layout evaluations. The best results highlighted in bold.

| Methods | Val | Ali | FD | DreamSim |
|---|---|---|---|---|
| AdaD2P | 0.8832 | 0.0923 | 33.55 | 0.1314 |
| Ours | **0.9765** | **0.0668** | **18.23** | **0.2436** |

## 5.4 RESULTS AND DISCUSSIONS

**Main results.** For mutlimodal content extraction, we present the experimental result in Table 3. Our MDE model outperforms the baselines due to its hierarchical structure, which is better suited

---

[3] https://huggingface.co/FacebookAI/roberta-base
[4] https://huggingface.co/facebook/bart-large

for handling long documents. In contrast, NeuralExt achieves the worst performance in image extraction, as it is a single-modal model that relies solely on figure and table captions to represent visual elements. We also conduct the ablation study to investigate the contribution of LSTM module in our MDE. From the Table 3, we can observe that our MDE suffers the significant decrease in performance when removing LSTM, which validates its effectivenss of LSTM module.

For multimodal generation, Table 4 compares the performance of our method against the baseline. Overall, our method demonstrates superior results. Specifically, while achieving comparable ROUGE scores for text output compared to AdaD2P, our method shows significant improvements in image output, with increases of 6.29% in ImgP and 6.81% in ImgR. This improvement is attributed to the effective multimodal representation modeling in our extractor. For layout output, our method achieves a 22.36% improvement in overlap and a 25.05% improvement in coverage. These gains can be attributed to the interactive generator, which employs an adaptive memory mechanism to capture long-term dependencies, and unifies both the paraphrasing of extracted content and layout generation, enhancing the interaction between content and layout. In order to evaluate the layout of generated poster holistically, we compute the metrics like validity (Val), alignment (Ali), the Frechet distance (FD) and DreamSim. The experimental results are shown in Table 5. Compared to baseline, our approach outperforms the baseline significantly.

**Ablation studies.** We also conduct ablation studies to assess the impact of different modules in the generator. The results, shown in Table 6, reveal that removing the memory module (i.e., f) leads to the largest decline in layout performance, underscoring the crucial role of the memory mechanism in maintaining and transforming state information. Additionally, configuration (a) outperforms (b), (c), (d), and (e), demonstrating the effectiveness of the KL-Divergence optimization, data extension strategy, pretraining strategy, and memory mechanism, respectively. Notably, compared to no memory or standard memory used in RMT (Bulatov et al., 2022), our adaptive memory design significantly improves layout generation, effectively managing the complexities of layout prediction.

Table 6: Overall result of different ablation settings under automatic evaluation metrics. "KL", "PT" and "DE" denotes KL-Divergence optimization, pretraining strategy and data extension strategy, respectively. The memory mechanism includes three variants: adaptive, normal and not used. All values in the table are expressed as percentages, with the best result highlighted in bold.

| | | | | | Text | | | Layout | |
| | KL | PT | DE | Memory | ROUGE-1 | ROUGE-2 | ROUGE-L | Overlap | Coverage |
|---|---|---|---|---|---|---|---|---|---|
| (a) | ✓ | ✓ | ✓ | Adaptive | **41.05** | **15.19** | **18.84** | 25.08 | **37.43** |
| (b) | ✓ | ✓ | ✗ | Adaptive | 40.26 | 14.53 | 18.47 | 38.85 | 20.06 |
| (c) | ✗ | ✓ | ✓ | Adaptive | 38.19 | 13.35 | 17.43 | 33.72 | 13.21 |
| (d) | ✓ | ✗ | ✓ | Adaptive | 38.90 | 13.79 | 17.85 | 30.91 | 17.68 |
| (e) | ✓ | ✗ | ✗ | Adaptive | 39.09 | 13.88 | 18.22 | **23.15** | 13.03 |
| (f) | ✓ | ✓ | ✓ | Normal | 40.12 | 14.60 | 18.29 | 36.42 | 23.42 |
| (g) | ✓ | ✓ | ✓ | None | 40.64 | 15.03 | 18.74 | 42.94 | 20.65 |

Table 7: Topic-aware evaluation results are obtained by training and testing on data from different conferences. For layout metrics, the values in parentheses represent the ground-truth posters. All values in the table are expressed as percentages, with the best results highlighted in bold.

| Topic | Text | | | Layout | | Image | |
| | ROUGE-1 | ROUGE-2 | ROUGE-L | Overlap | Coverage | ImgP | ImgR |
|---|---|---|---|---|---|---|---|
| All | **41.05** | **15.19** | **18.84** | **25.08** (5.11) | **37.43** (53.42) | 44.43 | 40.57 |
| CVPR | 26.33 | 11.61 | 13.38 | 34.01 (4.34) | 38.74 (59.14) | **50.01** | **66.58** |
| ICML | 36.74 | 13.84 | 15.42 | 32.11 (5.38) | 38.43 (50.28) | 34.19 | 56.43 |
| NeurIPS | 33.48 | 13.21 | 14.93 | 33.14 (5.07) | 38.26 (52.12) | 44.52 | 58.41 |
| ICLR | 34.81 | 13.15 | 15.51 | 32.45 (5.72) | 40.28 (53.47) | 29.88 | 55.29 |

**Topic-Aware Evaluation.** We evaluate performance in both topic-dependent and topic-independent settings. Specifically, we train and test our method on data from four conference proceedings: CVPR, ICML, NeurIPS, and ICLR. As shown in Table 7, the model trained on data from all topics outperforms models trained and tested within individual topics, particularly in terms of text and layout metrics. Notably, training on CVPR data yields the highest image extraction performance, with scores of 50.01% precision rate and 66.58% recall rate.

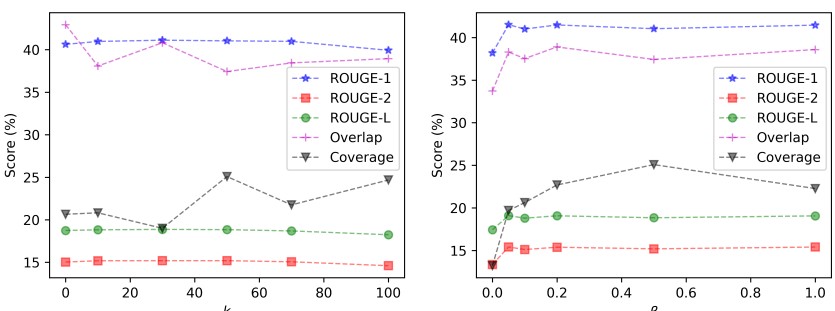

Figure 2: The analysis of parameter sensitivity for varying $k$ and $\beta$.

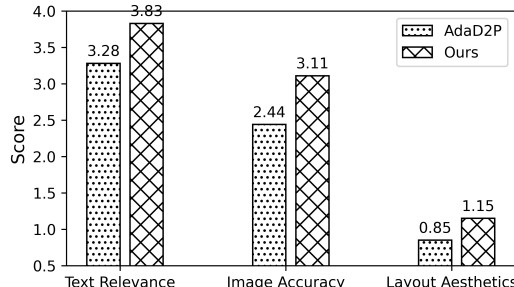

Figure 3: Human evaluation for the baseline and our model.

**Parameter Sensitivity.** We perform a parameter sensitivity analysis to examine the impact of memory size $k$ and the KL-Divergence weight $\beta$ on performance. To evaluate the effect of memory size, we experiment with values of 0, 10, 30, 50, 70, and 100. For the KL-Divergence weight, which balances the KL-Divergence loss and cross-entropy loss, we set values of 0, 0.05, 0.1, 0.2, 0.5, and 1.0. The results, displayed in Figure 2, indicate that the model achieves comparable ROUGE scores for textual metrics, while it performs best on layout metrics when the memory size is set to 50 and the KL-Divergence weight is 0.5. Given the challenges associated with layout prediction, we set the memory size ($k$) to 50 and the KL-Divergence weight ($\beta$) to 0.5 in our main experiments.

**Human Evaluation.** We conduct a human evaluation to assess the perceived quality of the generated posters. To simplify the task, we randomly sampled 50 document-poster pairs from the test dataset. For each document, we prepared three posters: the ground-truth poster, one generated by our method, and one produced by the baseline. We then ask ten annotators to rate these posters on a scale of 1 to 5 based on the following criteria:

- Text Relevance: How closely the text in the generated posters aligns with the content in the ground-truth poster.
- Image Accuracy: The accuracy of matching image elements between the generated and ground-truth posters.
- Layout Aesthetics: Whether the placement of text and image elements is both logical and visually appealing.

Figure 3 presents the average scores for each method. Our approach consistently receive higher ratings across all three aspects compared to the baseline. However, it is notable that both our method and the baseline received relatively low scores in the layout aesthetics category, highlighting the ongoing challenge of achieving aesthetic design in automatic poster generation systems. There remains significant room for improvement in this area.

## 6 CONCLUSION

In this paper, we introduced a new task, LayoutSciPG, for scientific poster generation. To tackle this, we first built a new dataset, namely SciPG, with over 10,000 pairs of scientific papers and their corresponding posters. We then developed a multimodal extractor to capture both text and image elements from the paper, and implemented an interactive generator with an adaptive memory mechanism to seamlessly integrate the paraphrasing of extracted content with layout generation. Both qualitative and quantitative evaluations highlight the effectiveness of our approach, while also revealing some remaining challenges.

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

## APPENDIX

### A1. DATASET PREPROCESSING

#### DATA PROCESSING

The collected scientific papers and poster pairs are in the PDF and PNG format, respectively. Data processing aims to extract the text and image elements and match them between documents and posters. However, it is not easy to exactly extract these multimodal elements, especially for images like tables and figures, from documents and posters because they contain rich structure and complex layouts. Except utilizing OCR tools to extract text information and its corresponding bounding box coordinates, it is also necessary to leverage the document layout analysis algorithms to obtain the layout information of figures.

**Paper document extraction.** In order to extract the image elements in the paper document, we employ a state-of-the-art (SOTA) document layout analysis model, i.e., VGT (Da et al., 2023), which are trained on the open sourced datasets such as Docbank (Li et al., 2020a) and Doclaynet (Pfitzmann et al.). Thus, we can obtain all the image elements in the document. For the text elements, we use an open-sourced OCR tool, i.e., PaddleOCR [5], to extract them.

**Poster extraction.** Compared to the paper document, a poster contain more complex and richer layout structure. Existing SOTA document layout analysis models like VGT perform not good on the poster. In addition, we also compare some commercial API service for document layout analysis. Finally, we select the best of them, i.e., Azure document layout analysis service [6], to extract text and image elements and their corresponding bounding boxes in the poster.

#### ELEMENT ALIGNMENT

The elements, i.e., text sentences and images, extracted from posters and documents need to be aligned each other. Thus, the aligned labels can be served as the supervised signals to guide which elements should be extracted from document and what to be generated according to the extracted elements. To this end, we employ a sentence matching model and an image matching model to achieve this:

**Sentence Matching.** We match sentences from the posters to their corresponding paper documents by using RoBERTa (Liu et al., 2019) to extract sentence embeddings from both. Matching sentences are identified by calculating cosine similarity between the embeddings.

**Image Matching.** For image elements, we match images from the posters to those in the corresponding document. We use a pre-trained CLIP (Radford et al., 2021) vision encoder to extract visual embeddings of all images in both the poster and the document, then match them based on the highest cosine similarity. Note that some images in posters may not appear in the corresponding document, resulting in no match. To simplify the process, we ignore any images whose highest visual embedding similarity is below a threshold of $\delta = 0.8$.

### A2. QUALITATIVE RESULTS

Figure 4 presents qualitative results for scientific poster generation. Our method first extracts and paraphrases the text and image elements, followed by predicting the corresponding layout boxes. We

---

[5] https://github.com/PaddlePaddle/PaddleOCR
[6] https://azure.microsoft.com/en-us/products/ai-services/ai-document-intelligence

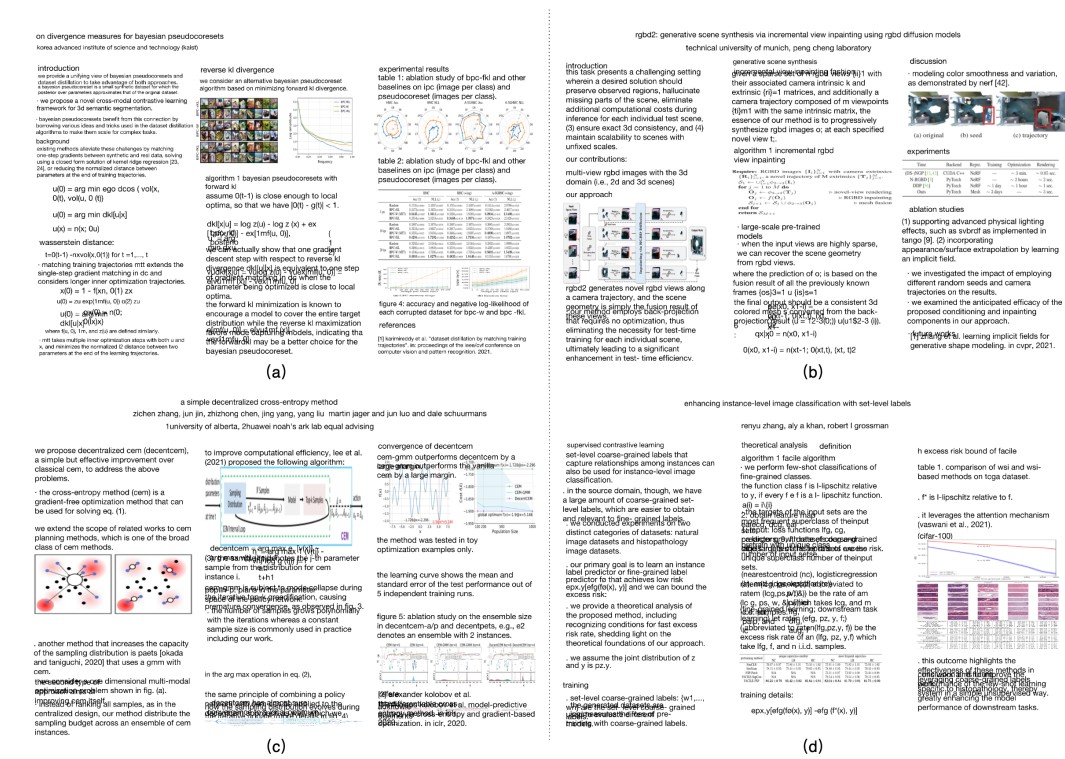

Figure 4: Four generated posters by the proposed method.

then utilize the Python package *pptx* [7] to generate editable posters. For simplicity, font size, style, and capitalization of input text are disregarded during the automatic generation process. While the generated posters appear visually acceptable at first glance, several challenges remain. First, elements in the poster often overlap. Second, some areas of the canvas are underutilized. Third, predicted boxes sometimes fail to accommodate the associated text elements. Most importantly, the logical relationships between these elements may be misaligned. To solve these problems, this work aims to establish a benchmark dataset and approach to advance the development of automatic poster generation.

---

[7] https://pypi.org/project/python-pptx

