# OpenReview forum: "SciPG: A New Benchmark and Approach for Layout-aware Scientific Poster Generation"
_ICLR.cc/2025/Conference — ICLR 2025 Conference Withdrawn Submission_

### Official Review · Reviewer_hLn1 · 2024-10-24

**Soundness:** 3
**Presentation:** 3
**Contribution:** 2
**Rating:** 3
**Confidence:** 4

**Summary:**

This work studies an important problem – automatically generate a poster from a scientific paper. It contributes a dataset with 10k pairs of scientific papers and their corresponding posters. It proposes an extractor to retrieve critical text and image elements from the papers and a generator to paraphrase the extracted content and generate a layout.

**Strengths:**

1.	It focuses on a very important and practical problem – scientific poster generation. This problem receives little attention currently. If it is well-solved, lots of people will benefit from it.

2.	The paper is easy to follow. The idea is clearly expressed.

3.	It contributes a new dataset for scientific poster generation, which will have big impact on this domain if released.

**Weaknesses:**

1.	The reason behind the model architecture choice is not clearly explained.

a) Why is BiLSTM instead of Transformer used in multimodal extractor? The other components in the framework are all based on Transformer. What is the reason for such a special design that only uses BiLSTM in this part?

b) In the interactive generator, what does ‘interactive’ refer to here? Does it mean users can participate in this process?

2.	The evaluation for the layout part is not sufficient. Layout is important for a scientific poster as it relates to whether the poster is attractive and conveys information effectively.

a) There are lots of studies focus on the layout generation problem [1][2]. They take the elements as input and generate their positions.  The comparison with them could be using the output of the multimodal extractor as their input. If the performance of the existing layout generation model is worse than the proposed interactive generator, I will be convinced that it is necessary to design a specific interactive generator; otherwise, the novelty and contribution of this work will be very limited.

b) Important metrics for layout are missing, e.g., FID and alignment used in existing work [1][2].

3.	There is no discussion and ablation study about whether the problem decomposition is reasonable. Currently, the problem is decomposed into two parts, where the first part is to extract key text and image from the paper and the second part is to paraphrase text as well as generate layout. Why not put the paraphrase task a separate part or merge it into the first part, since it is a natural language processing task and is far from layout generation problem? Besides, if the task is decomposed as I suggested, the existing techniques [1][2] about layout generation can be reused, which may be beneficial for overall performance.

4.	There are only two qualitative results, which are not enough for justifying the performance of the proposed method. Besides, from the qualitative results shown, I can find many overlaps between elements, which indicate that the performance is not good enough.

[1] LayoutFormer++: Conditional Graphic Layout Generation via Constraint Serialization and Decoding Space Restriction

[2] PosterLlama: Bridging Design Ability of Language Model to Contents-Aware Layout Generation

**Questions:**

For the discussion in Section 4.3.2, what is the total token length for this task?

---

> ### Author Response · Authors · 2024-11-22
>
> Q1-a：Why design a BiLSTM layer？
>
> R1-a: As shown in Table 2 of our paper, the input text is quite lengthy, with an average of over 10,000 tokens. However, a standard Transformer model can only process up to 512 tokens at a time. To address this limitation, we adopt a hierarchical framework for the semantic representations. In this framework, a Transformer model at the lower level learns sentence-level semantic representations, while a BiLSTM at the higher level captures the contextual semantic representations for the entire document. We have clarified this in our revised version.
>
> In addition,  we also investigate the contribution of LSTM in the ablation study. The experimental results are shown in the Table.
> |  Methods   | ROUGE-1 | ROUGE-2 | ROUGE-L|  ImgP | ImgR  |
> |  ----           | ----          | ----           | ----          | ----   | ----     |
> | MDE           |     40.68   | 14.76        | 17.54        | 44.43 | 40.57 |
> | MDE w/o LSTM |     36.73   | 11.78        | 14.59        | 40.26 | 36.55 |
>
> From the experimental results, we can observe that our MDE suffers the significant decrease in performance when removing LSTM. We have updated the experimental results in our uploaded revised version of our paper.
>
> Q1-b：what does ‘interactive’ refer to here?
>
> R1-b:  Sorry for your confusions. Our ‘interactive’ process does not involve user participation. It refers to an iterative generation process, as opposed to generating all content at once. The entire process is similar to dialogue generation. Given an input list from the extractor, at each step, we automatically feed one element of the input into the generator, which then returns its predicted result. This process is repeated iteratively until the entire input list has been processed.
> The design motivation behind this approach is driven by two main factors:
> 1）The input list contains a large amount of text, and concatenating all of it would result in a lengthy input that the generator cannot process at once.
> 2）By feeding the input interactively, the generator can better learn the relationship between the current element and previously predicted elements, which enhances both layout prediction and paraphrasing content generation.
> We have clarified this in our revised version.
>
> Q2-a:  The layout generation problem.
>
> R2-a: Before we present our work, we have carefully examined the existing layout generation task, which differs significantly from the layout prediction subtask in scientific poster generation. The layout generation task typically involves predicting the position and category of graphic elements on an empty canvas, an image (e.g., advertising posters), or an image with text constraints. However, existing layout generation methods primarily focus on the element categories, generating layouts with different categories of elements without considering the actual textual or visual content within each element. Essentially, these methods create layout templates with different categories of bounding boxes, which can result in a misalignment between the predicted bounding boxes and the content inside them. For example, consider a case with 20 elements, including 15 text elements and 5 figures. While existing methods might generate 20 bounding boxes with visually coherent layouts based on element categories (e.g., text and figures), they fail to address which specific text or figure should be placed in which bounding box.
>
> In contrast, in our scientific posters, each element contains not only category information but also specific semantic content, which is crucial for ensuring the semantic consistency and coherence of the poster. Existing layout generation methods are not well-suited to our scenario. To resolve this, we propose an interactive layout prediction approach that focuses not only on the element category but also on the semantic content within each element.
> Given the complexity and diversity of scientific poster layouts, constructing a new dataset based solely on our collection of scientific posters could be valuable for the existing layout generation task. We also use the current SOTA layout generation approach, namely LayoutDM[1], to generate the scientific poster template based on our dataset. Unfortunately, LayoutDM[1] performs not good, generating plenty of overlapped bounding boxes. However, this task would be a distinct task from our own scientific poster generation. We plan to further explore this direction in future work.
>
> [1] LayoutDM: Discrete Diffusion Model for Controllable Layout Generation

---

> > ### Author Response · Authors · 2024-11-22
> >
> > Q2-b: Important metrics for layout are missing.
> >
> > R2-b: As discussed above, the existing layout generation problem differs significantly from our task. Consequently, not all layout evaluation metrics in the layout generation task are suitable for our scenario. We select the appropriate metrics, such as Validity, Alignment and FD, for the evaluation of our scientific poster generation. Among them, Validity, annotated as Val, is the ratio of valid elements greater than 0.1% of the canvas. Alignment, annotated as Ali, is the extent of spatial non-alignment between elements.  FD denotes the Frechet distance. The experimental results are shown in following table.
> > |  Methods   |       Val     |        Ali      |       FD      |
> > |  ----           | ----          | ----          | ----          |
> > | AdaD2P     |    0.8832  | 0.0923    |  33.55     |
> > | ours           |     0.9765    | 0.0668   | 18.23      |
> > We have updated the experimental results in our uploaded revised version of our paper.
> >
> >
> >
> > Q3:  The problem decomposition. if the task is decomposed as I suggested, the existing techniques about layout generation can be reused,
> >
> > R3: Thank you for your suggestions. As discussed earlier, even though we divide our task into three subtasks: content extraction, content paraphrasing, and layout prediction, the existing layout generation techniques cannot be directly applied to our layout prediction subtask. Essentially, they only create layout templates with different categories of bounding boxes while ignoring the semantic information within each element.
> >
> >  In fact, our baseline model, AdaD2P, follows a three-phase (i.e., content extraction, content paraphrasing, and layout prediction) approach for poster generation. However, our proposed method unifies content paraphrasing and layout prediction into a single subtask to better capture both the semantic representation within each element and the relationships among elements.
> >
> >
> >
> > Q4: Overlaps between elements problem.
> >
> > R4: We acknowledge that the proposed approach is still a step away from fully automated poster generation. Even state-of-the-art AI models, such as ChatGPT, can make errors when performing specific tasks. However, our method focuses on automatically generating editable draft posters from research papers, enabling researchers to make only minor adjustments to produce the final poster, thereby saving significant time.
> >
> > Scientific poster generation is inherently challenging, with several complexities, including diverse and intricate layouts, a large number of elements, multimodal processing, and handling lengthy inputs. To address these challenges, we aim to release a new dataset and introduce a baseline approach to advance research in the field of scientific poster generation.
> >
> >
> > Q5: The total token length.
> >
> > R5: Sorry for your confusions. The total token length in Section 4.3.2 represents the length of the entire tokenized contextual memory, as shown in Figure 1. In this memory, each element consists of a tuple containing the target output and the flattened layout information.

---

> > > ### Comment · Reviewer_hLn1 · 2024-11-25
> > >
> > > R3: In my original feedback, I suggest a comparison between different problem decomposition, including 1) a three-stage framework, where content extraction, content paraphrase and layout generation are separate stages ("put the paraphrase task a separate part" in my original feedback); 2) a two-stage framework, where content extraction and content paraphrase is put in the first stage and layout generation are put in the second stage ("merge it into the first part, since it is a natural language processing task and is far from layout generation problem" in my original feedback).
> > > The second item is not addressed.

---

> > > > ### Author Response · Authors · 2024-11-25
> > > >
> > > > R3. Thank you for your suggestions. Our scientific poster generation can be structured into different frameworks:
> > > > 1.Ext → Par → Lay: A three-stage framework where content extraction, content paraphrasing, and layout generation are handled independently.
> > > > 2.Ext + Par → Lay: A two-stage framework where content extraction and paraphrasing are merged (similar to a summarizer), followed by layout prediction for the generated content.
> > > > 3.Ext → Par + Lay: Another two-stage framework where content paraphrasing and layout generation are combined. Our proposed method adopts this two-stage framework.
> > > > The experimental results for these frameworks are shown in the table below.
> > > >
> > > > |  Methods   | ROUGE-1 | ROUGE-2 | ROUGE-L|  ImgP | ImgR  | Overlap | Coverage|
> > > > |  ----           | ----          | ----           | ----          | ----   | ----     |  ----   | ----     |
> > > > | Ext->Par->Lay |     40.23   | 14.38       | 17.03       | 44.43 | 40.57 | 23.53  |  35.76  |
> > > > | Ext+Par->Lay   |    41.36   | 15.72        | 18.96      | 44.57 | 40.62   | 24.21  |  36.24  |
> > > > | Ext->Par+Lay   |     41.05   | 15.19        | 18.84       | 44.43 | 40.57 | 25.08  |  37.43  |
> > > >
> > > > From the results, we observe that Ext → Par → Lay achieves the worst performance in both text and layout metrics compared to the two-stage frameworks. A potential reason is that treating the tasks as isolated subtasks introduces greater error propagation, leading to a performance drop.
> > > > When comparing Ext + Par → Lay and Ext → Par + Lay, the former merges content extraction and paraphrasing, which improves text and image metrics but slightly decreases layout performance. This could be attributed to the enhanced representation learning ability enabled by multi-task learning (extraction and paraphrasing).
> > > > Similarly, in Ext → Par + Lay, combining paraphrasing with layout generation improves layout prediction performance, highlighting the benefits of joint optimization.
> > > >
> > > > We will update the experimental results in our revised manuscript.

---

> > ### Comment · Reviewer_hLn1 · 2024-11-25
> >
> > R1-a: Could you explain the reason behind or point out references to the claim that " a standard Transformer model can only process up to 512 tokens at a time". Llama [1] and Phi [2] are also standard Transformer model. The token limit is not 512 for them.
> >
> >
> > R2-a: In your example, if there are 20 elements, including 15 text elements and 5 figures, existing method can generate the bbox binding to the specific element. For example, PosterLlama [3] can achieve it. Besides, PosterLlama also considers the element content as the context for layout generation instead of only category information. Similarly, Graphist [4] can also address these problems. The methods they proposed can also be applied to scientific posters.
> >
> > [1] https://arxiv.org/pdf/2407.21783
> >
> > [2] https://arxiv.org/pdf/2404.14219
> >
> > [3] https://arxiv.org/pdf/2404.00995
> >
> > [4] https://arxiv.org/pdf/2404.14368

---

> > > ### Author Response · Authors · 2024-11-25
> > >
> > > R1-a: In our extractor, to better capture text semantic representations, we typically utilize a pretrained language model like BERT or RoBERTa. These models employ a standard Transformer encoder architecture and are pretrained on large-scale corpora. During pretraining, these models are trained with a context token length of 512. Using the same setup allows our encoder to inherit the semantic knowledge from the pretrained model effectively.
> > > In contrast, models like LLaMA and Phi adopt a Transformer decoder architecture, which is more suited for language generation rather than representation learning. These models are pretrained with significantly longer context lengths, resulting in substantial GPU memory requirements. For instance, LLaMA 3 processes information in a context window of up to 128K tokens, which demands extensive computational resources. We will clarify this in our revised version of our paper.
> > >
> > >
> > >
> > > R2-a：I am confident that PosterLlama cannot effectively handle the semantic content within each element. A key limitation is that the constructed input in PosterLlama does not consider and process visual semantic features, even when visual elements are present. Visual elements such as figures, charts, and tables are common in scientific poster generation, yet PosterLlama lacks the capability to integrate their semantic information.
> > > Even for text elements, PosterLlama introduces only a single field, $Text \  Constraint$, to describe text content, element relationships, or attributes for all elements. However, these descriptions are not aligned with their corresponding elements. Additionally, the $Text \  Constraint$ field is designed for advertisement layout generation, which typically involves short and simple descriptions. In contrast, scientific poster generation often includes several dozens of long sentences. Using such a design for Text Constraint makes it difficult for the model to determine which content belongs to which text element, leading to disordered text elements. Furthermore, as noted in the limitations of PosterLlama (Page 12), it struggles to account for text length in the generated layout. However, in scientific poster generation, the order and length of bounding boxes for text elements are critical.
> > >
> > > Regarding your recommendation of Graphist, it represents element content by treating each element as an RGBA image and using a visual encoder to extract semantic features. While this approach works well for advertisement poster layouts, where text elements are short and often defined by specific shapes or appearances, it is inadequate for scientific posters. In our case, text elements consist of long sentences, and their relationships are crucial for layout coherence. Using only a visual encoder to represent text elements fails to capture the semantic information and relationships necessary for scientific poster generation.

---

> ### Comment · Reviewer_hLn1 · 2024-11-25
>
> R1-a:
>
> The reason for choosing BiLSTM instead of Transformer is not convincing.
>
> 1) The Bart-like [1] model uses a Transformer Encoder and a Transformer Decoder, which is a very successful architecture across many tasks. The difference between the model proposed in this work and Bart-like model is that this work uses a BiLSTM instead of a Transformer Decoder. There should be reasons for not using Transformer Decoder like Bart. Note that I do not mean using the exact architecture as Bart. I mean the Transformer Decoder like Bart is also a potential choice.
>
> 2) The correct way of using Llama and Phi in the task proposed in this work is not using them as a representation learning method. Instead, they can be finetuned to directly predict the extractive score. First, as these models are pretrained with longer context and more data, they may provide stronger prior knowledge and lead to better performance. Second, this approach eliminates the need for an extra BiLSTM and training from scratch.
>
> In summary, Bart-like architecture (Transformer Encoder and Decoder) and Llama and Phi-like architecture (Transformer Decoder) are powerful choices. Given their strong performance across many tasks, the use of a BiLSTM in this work requires further explanation and comparison with these models.
>
> In your current feedback, do you mean BiLSTM is better at processing longer sequence than Transformer Decoder? Could you please provide related work as well as experimental results to demonstrate it?
>
> [1] https://arxiv.org/abs/1910.13461
>
> R2-a:
>
> 1) semantic content or visual semantic features.
>
> PosterLlama uses DINO v2 and Adapter to encode the semantic content of visual elements (see Section 3.2 of PosterLlama).
>
> 2) which content belongs to which text element.
>
> PosterLlama can achieve it with a very minimal revision in the prompt. As shown in Appendix F of PosterLlama, by putting text content in each <rect /> item, it can achieve which content belongs to which text element.
>
> 3) long sentences problem.
>
> How is this problem addressed by the approach in this work? I observed lots of overlaps between textboxes in qualitive results.

---

> > ### Author Response · Authors · 2024-11-25
> >
> > R1-a:
> >
> > 1.The Transformer Decoder in BART is not a suitable choice for our multimodal extractor, as it is designed primarily for generative tasks such as machine translation. Our extractor focuses on identifying and extracting important sentences or figures from a paper. To achieve this, we use a hierarchical structure (RoBERTa + BiLSTM) to capture the contextual representation of each element (e.g., sentences, figures, or tables) and then predict whether each element should be selected based on the learned representation.
> > The Transformer Decoder, being an autoregressive model, is typically used for sequence generation rather than representation tasks. Its unidirectional nature prevents it from effectively capturing the contextual semantic representation required for our task.
> >
> >
> > 2. Thanks for your suggestions. LLaMA- and Phi-like architectures, based on Transformer Decoders, are autoregressive models. These models are not designed to provide representations for sentences or figures, making them unsuitable for predicting extractive scores directly. Attempting to fine-tune such decoder-based models for this purpose is fundamentally impractical. Instead, it is more reasonable to fine-tune these models for tasks like content paraphrasing, similar to a summarization task. However, this approach introduces two significant challenges: handling multimodal elements and managing the substantial GPU memory requirements for processing long documents. As far as we know, there are no effective training approaches for directly summarizing multimodal long documents using such models.
> > In our scenario, the BiLSTM component could be replaced by a two-layer Transformer Encoder for representation learning, but not by a Transformer Decoder. The Transformer Encoder utilizes multi-head attention to capture global contextual representations, while the Transformer Decoder relies on causal attention, which cannot effectively model global context.
> > Thus, the suitability of BiLSTM for processing longer sequences is not a direct limitation of Transformer Decoders. Instead, the choice of architecture depends on the need for global contextual representation, which Transformer Decoders inherently lack.
> >
> > R2-a:
> >
> > 1. It seems there is a misunderstanding regarding PosterLlama. It employs DINO v2 and Adapters to encode the advertisement background image rather than visual elements like figures and tables, which are integral to our scientific posters. Additionally, PosterLlama is designed to process only a single background image (as shown in Fig. 1 on Page 4), whereas our scenario involves handling multiple figures and tables.
> >
> > 2. As you mentioned, PosterLlama can indeed achieve this with minimal modifications to the prompt. By embedding the text content directly within each <rect /> item, it can establish which content corresponds to which text element. This approach is quite similar to our proposed method, where we concatenate the specific content of each element with masked coordinate information. The primary difference lies in the format—we do not use HTML.
> > However, if there are several dozen elements, embedding text content in each <rect /> item would result in a very long input sequence (potentially spanning several thousand tokens). This would significantly increase GPU memory requirements. To address this challenge, we adopt an interactive training approach, which is better suited to handling such extensive inputs efficiently.
> >
> > 3. Given an input list, each sentence is encoded in an interactive mode, conditioned on the previous memory, until the end of the list. This approach effectively reduces GPU memory usage. Existing methods cannot fully prevent overlap issues. As noted in the limitations of PosterLlama (Page 12), it struggles to account for text length in the generated layout, which can also lead to overlap problems in our scenario.

---

> > > ### Author Response · Authors · 2024-11-27
> > >
> > > Dear Reviewer,
> > >
> > > I am confident that existing layout generation approaches, such as PosterLlama, cannot effectively handle the multimodal elements present in scientific posters, making them unsuitable for our layout prediction subtask. Furthermore, LLaMA- and Phi-like architectures, which are based on Transformer decoders, are also ill-suited for representing multimodal elements in paper documents.
> > >
> > > I look forward to your feedback. Thank you.

---

> > > > ### Author Response · Authors · 2024-11-29
> > > >
> > > > Dear reviewer,
> > > >
> > > > We sincerely appreciate your time and effort in reviewing our manuscript and offering valuable suggestions. As the author-reviewer discussion phase is drawing to a close, we would like to confirm whether our responses have effectively addressed your concerns. If you require further clarification, please do not hesitate to contact us.
> > > >
> > > > Best regards.

---

> > > > > ### Comment · Reviewer_hLn1 · 2024-12-01
> > > > >
> > > > > Thanks for your feedback. I would like to keep my score – a clear rejection due to the concerns about technical solidity.
> > > > >
> > > > > 1.	The proposed problem decomposition is not optimal.
> > > > >
> > > > > In the discussion, there is an added ablation study about problem decomposition, including Ext->Par->Lay, Ext+Par->Lay and Ext->Par+Lay. The results show that over seven metrics, Ext+Par->Lay (the one not used in this work) performs better than Ext->Par+Lay (the one proposed in this work) for six metrics.
> > > > >
> > > > > 2.	The reason for using a BiLSTM upon a Transformer Encoder is not convincing.
> > > > >
> > > > > Bart-like or Llama/Phi-like architecture can be used by taking the document as input and generating the chosen sentences (or extractive scores) as output. The image features can be handled by Llama/Phi vision version or Llava. As mentioned in my former feedback, since Bart-like and Llama/Phi-like architectures are more widely used than Transformer Encoder+BiLSTM architecture in the literature and they have demonstrated strong performance across many tasks, the use of a BiLSTM in this work requires further explanation and comparison with these models.
> > > > >
> > > > > 3.	Baselines for layout generation are missing.
> > > > >
> > > > > I acknowledge that the concrete tasks are different, and the baselines need some adaptation to be applied to the task in this work. However, the proposed method often generates layouts with severe overlaps (shown by qualitative results) while the baseline methods seldom generate layouts with overlaps. If there is no direct comparison between the proposed method and the baselines, we can never know the reason behind – 1) the task in this work is too difficult and baseline methods also generate lots of overlaps on this task or 2) the proposed method is not good enough.

---

> > > > > > ### Author Response · Authors · 2024-12-02
> > > > > >
> > > > > > R1: Our model adopts the Ext->Par+Lay architecture instead of Ext+Par->Lay because Ext->Par+Lay achieves better performance in layout metrics, which we consider more critical for scientific poster generation.
> > > > > >
> > > > > > R2: We utilized a typical hierarchical architecture for multimodal information extraction from long documents. Do you fully understand the capabilities and limitations of decoder-only models?  Bart-like or Llama/Phi-like architectures are better suited for text generation tasks. Moreover, treating our extraction task as a generation task would result in excessively long input and output sequences, making it intractable and impractical to train Llama/Phi-like models. Additionally, we directly experimented with GPT-4o (architecturally similar to Llama/Phi) for multimodal content generation, but its performance on our task was unsatisfactory.
> > > > > >
> > > > > > R3: Your statement, “the baseline methods seldom generate layouts with overlaps,” is incorrect. We adapted our scientific poster dataset to the existing layout generation task and applied a state-of-the-art layout generation model, LayoutDM[1]. The results were suboptimal, with significant overlaps and low coverage.
> > > > > >
> > > > > > It is essential to recognize that our proposed task is highly challenging. Our goal is to release a new dataset and present a baseline approach to advance research in the field of scientific poster generation.
> > > > > >
> > > > > >
> > > > > > [1] LayoutDM: Discrete Diffusion Model for Controllable Layout Generation

---

### Official Review · Reviewer_93YC · 2024-11-02

**Soundness:** 3
**Presentation:** 3
**Contribution:** 3
**Rating:** 8
**Confidence:** 3

**Summary:**

The authors define the task of layout-aware scientific poster generation (LayoutSciPG) which takes both content extraction and layout into account (compared to previous work which looks at these things in isolation). To facilitate data-driven approaches the authors first collect a novel dataset (SciPG) of paper-poster pairs and automatically align text and image contents. The authors then introduce a novel two-stage pipeline architecture tailored to LayoutSciPG and finetune it on SciPG). Both in automatic and human evaluation the authors showcase the effectiveness of their approach to automatically generate scientific posters conditioned on papers.

**Strengths:**

The provided dataset is orders of magnitudes larger then existing related datasets and will surely be useful to other researchers. The authors promise to release code and data artifacts.

The proposed architecture is also very novel and constitutes a core contribution of this work. Even though there are a lot of newly introduced custom components, the authors quantify the contribution of each component in an ablation study.

The human evaluation  supports the findings from automatic evaluation and increases their credibilty.

**Weaknesses:**

The main experimental results are a bit lacking due to the lack of baselines (only one baseline provided). While I understand that the task is a novel one, I don't see why the approaches that tackle content extraction and layout in isolation mentioned in the introduction and related work couldn't serve as baselines. Providing a simple end2end baseline (or maybe a diffusion-based one) would also have been insightful.

The paper definitely needs more examples. As of know only two (bit hard to read) examples are provided in the main text. The authors should provide more examples in the appendix.

While the paper is generally easy to follow some parts don't feel very polished and can potentially be confusing. E.g., there are are passages with duplicated information (l. 45-51 & l. 125-130) and sometimes the paper is short on details (l. 387-388 & l. 395-397).

**Questions:**

#### Comments

The title of the paper is SciPG, but this term is not defined anywhere in the text. Only from Table 1 do we know that this is the name of the dataset. This leads to confusion especially since a similar term (LayoutSciPG) is used very often.

ImgR and ImgP are not clearly defined. While it can be understood from the context, being more precise could alleviate confusion.

Table 2 is a bit hard to read. Which column belongs to documents and which one to posters? Adding vertical bars or cmidrule's could help.

#### Questions

One thing I don't understand is the role of the BiLSTM on top of the Roberta embeddings. The authors claim that this "captures contextualized representations" (l.257) but the Roberta embeddings should already be contextualized. Unfortunately the contribution of the LSTM has not been investigated in the ablation study.

Posters might contain original content (e.g. images that do not appear in the paper as mentioned in l.183). Do the authors have an idea how that could be addressed in future work?

Your dataset contains a validation split but do you actually use it somewhere? Maybe for the experiments in Figure 2?

Do the authors think that a perceptual image similarity metric [1] between generated and reference posters could be a useful addition to the automatic evaluation?

[1] DreamSim: Learning New Dimensions of Human Visual Similarity using Synthetic Data

---

> ### Author Response · Authors · 2024-11-22
>
> Q1: Why the approaches that tackle content extraction and layout in isolation mentioned in the introduction and related work couldn't serve as baselines. Providing a simple end2end baseline would also have been insightful.
>
> R1: Sorry for your confusions. As a novel task, there are few feasible approaches that can be directly adapted to our scenario. Previous research on scientific poster generation has primarily focused on either poster composition or content extraction. The former proposed a simple probabilistic graphical model to infer panel attributes, but it requires human annotation for the panels of a poster. In contrast, our new dataset does not include panel information.
> For content extraction, we compare our model against two additional baselines: NeuralExt [1] and MSMO [2]. NeuralExt is a neural extractive model designed to extract text, figures, and tables from a paper. MSMO is a multimodal attention model that jointly generates text and selects the most relevant image from multimodal input. We adapted MSMO for the content extraction task. The experimental results are presented in the following table.
> |  Methods   | ROUGE-1 | ROUGE-2 | ROUGE-L|  ImgP | ImgR  |
> |  ----           | ----          | ----           | ----          | ----   | ----     |
> | NeuralExt   |     36.55   | 12.67       |  14.43      | 31.68 | 24.91 |
> | MSMO        |     32.45   | 10.43       |  12.51      | 36.64 | 32.56 |
> | AdaD2P      |     38.28   | 13.04       |  15.72      | 38.24 | 33.76 |
> | ours           |     40.68   | 14.76        | 17.54        | 44.43 | 40.57 |
>
>
> Our model outperforms the baselines due to its hierarchical structure, which is better suited for handling long documents. In contrast, NeuralExt achieves the worst performance in image extraction, as it is a single-modal model that relies solely on figure and table captions to represent visual elements.
>
> Regarding an end-to-end baseline, there are currently no feasible end-to-end approaches that can be adapted to our task. Jointly training a content extraction model and a layout generation model is intractable due to the challenges posed by handling multimodal long documents and generating complex layouts.
>
> We have updated the experimental result of multimodal content extraction in our uploaded revised version of paper.
>
> [1] Xu et al. Neural Content Extraction for Poster Generation of Scientific Papers
> [2] Zhu et al. MSMO: Multimodal Summarization with Multimodal Output.
>
>
>
> Q2: Providing more examples in the appendix, content polishment and the term SciPG is not defined anywhere in the text.
>
> R2: Thanks for your suggestions. More examples, the duplicated information and the lacked details on evaluation metrics  have been clarified in our revised version of our paper.  For SciPG, it denotes our benchmark. We have clarified its definition in our introduction.
>
>
> Q3: ImgR and ImgP are not clearly defined.
>
> R3:  Sorry for your confusions. ImgR represents the recall of extracted image elements, calculated as $\frac{\text{The number of correct images}}{\text{The total number of ground truth images}}$. ImgP represents the precision of extracted image elements, calculated as $\frac{\text{The number of correct images}}{\text{The total number of extracted images}}$. We have clarified this in the revised version of our paper.
>
> Q4: Adding vertical bars or cmidrule's in Table 2.
>
> R4: Thanks for your suggestions. We have updated Table 2 in the revised version.
>
> Q5: The BiLSTM on top of the Roberta embeddings.
>
> R5: Sorry for your confusion. As shown in Table 2 of our paper, the input text is quite lengthy, with an average of over 10,000 tokens. However, a standard Transformer model (Roberta) can only process up to 512 tokens at a time. To address this limitation, we adopt a hierarchical framework for the semantic representations. In this framework, a Transformer model at the bottom level learns sentence-level semantic representations, while a BiLSTM at the higher level captures the contextual semantic representations for the entire document.
>
> In addition,  we also investigate the contribution of LSTM in the ablation study. The experimental results are shown in the Table.
> |  Methods   | ROUGE-1 | ROUGE-2 | ROUGE-L|  ImgP | ImgR  |
> |  ----           | ----          | ----           | ----          | ----   | ----     |
> | MDE           |     40.68   | 14.76        | 17.54        | 44.43 | 40.57 |
> | MDE w/o LSTM |     36.73   | 11.78        | 14.59        | 40.26 | 36.55 |
>
> From the experimental results, we can observe that our MDE suffers the significant decrease in performance when removing LSTM. We have updated the experimental results in our uploaded revised version of our paper.

---

> > ### Author Response · Authors · 2024-11-22
> >
> > Q6: Do the authors have an idea how that could be addressed the images that do not appear in the paper in future work?
> >
> > R6: In our work, we assume that the image elements (figures and tables) in the poster are derived from the corresponding paper document. While there may be a very small number of unmatched images between the posters and papers, this issue could potentially be addressed using image generators, such as diffusion models. However, generating plausible scientific figures remains a significant challenge.
> >
> >
> > Q7: The validation dataset.
> >
> > R7: Our model is trained on our training dataset. During training, we iteratively optimize the model parameters. Every $T$ iterations, we evaluate the model's performance on the validation dataset using the current parameters. After completing the training process, we load the model with the optimal parameters, as determined on the validation dataset, and test it on the test dataset. We have clarified this in the revised version of our paper.
> >
> > Q8: The problem of perceptual image similarity metric
> >
> > R8: Thanks for your suggestions. DreamSim, a perceptual image similarity metric, is primarily designed for natural images. Posters, however, are characterized by text, scientific figures, and tables, which differ significantly from natural images. Nevertheless, we evaluated the DreamSim metric between the generated and reference posters. The experimental results are shown as below:
> > |  Methods   | DreamSim |
> > |  ----           |   ----          |
> > | AdaD2P     |     0.1314       |
> > | ours           |    0.2436         |
> >
> > We have updated the experimental results in our revised version of our paper.

---

> ### Comment · Reviewer_93YC · 2024-11-24
>
> I thank the authors for their response. The authors were able to address my points of concern and thus I am raising my scores. I still recommend increasing the sizes of the example posters in the appendix, perhaps by putting them on separate pages.

---

> > ### Author Response · Authors · 2024-11-25
> >
> > Thanks for your comments and suggestions. We will provide more example posters in the appendix of our revised manuscript.

---

### Official Review · Reviewer_g553 · 2024-11-03

**Soundness:** 3
**Presentation:** 3
**Contribution:** 3
**Rating:** 6
**Confidence:** 3

**Summary:**

This paper investigates the layout-aware scientific poster generation (LayoutSciPG) task. Specifically, it addresses three challenges: multimodal extraction, multimodal generation, and the need for large-scale training data. To tackle LayoutSciPG, the authors develop a multimodal extractor-generator framework that includes extraction and interactive generation modules. Overall, the proposed solution is sound and reasonable. Additionally, a novel poster dataset has been constructed.

**Strengths:**

1.	This work explores an interesting task called layout-aware scientific poster generation, which is useful for generating flexible posters from scientific papers through integrated automatic content extraction and layout design.
2.	A large-scale dataset is created, containing over 10,000 pairs of scientific papers and their corresponding posters.
3.	Extensive experiments are conducted to evaluate both the qualitative and quantitative performance of the proposed approach.
4.	Practical issues, such as GPU memory consumption and long-term dependencies, are considered and addressed in this paper.

**Weaknesses:**

1.	Although the paper claims to present a novel research task, the research novelty is not particularly significant, as poster design has been extensively studied.
2.	The technical contributions are marginal, as most techniques have been developed and are commonly used. For instance, the multimodal extractor (MDE) is based on RoBERTa and BiLSTM, while the interactive generator (IG) relies on BART and RMT. The developed framework is relatively straightforward. The authors could better highlight their differences to better highlight its technical novelties.
3.	The experimental results are not convincing enough, as they only compare with one baseline, AdaD2P. Including comparisons with more recent advanced baselines would strengthen the advantages and make the empirical results more persuasive.
4.	As shown in Figure 4, the generated posters are still poor and not suitable for practical applications. It would be beneficial to present and compare posters generated with other baselines.

**Questions:**

Is it possible to incorporate prompts to generate the posters, allowing for personalized control by users over the final output?

---

> ### Author Response · Authors · 2024-11-22
>
> Q1: The research novelty is not particularly significant, as poster design has been extensively studied.
>
> R1: To the best of our knowledge, automatic scientific poster generation remains an underexplored area. Existing layout generation tasks primarily focus on advertisement poster design for products, where the diversity and complexity of layouts are significantly lower than in our case. In contrast, for scientific posters, previous studies have largely focused on either content extraction or poster composition, often relying on small-scale datasets. The lack of large, publicly available datasets has further hindered progress in this field.
> Poster composition methods have typically employed simple probabilistic graphical models to infer panel attributes, but these approaches require human annotation of poster panels. On the other hand, content extraction methods rely on predefined templates to generate fixed-format posters, which limits both the flexibility and aesthetic appeal of the resulting scientific posters.
> To address these challenges, we are the first to construct a large-scale dataset and propose a layout-aware poster generation approach that enables more flexible and visually appealing poster designs.
>
>
> Q2:The authors could better highlight the differences of our approach.
>
> R2: The main contributions of our work are the release of a new dataset and the introduction of a baseline approach to advance research in scientific poster generation. In our proposed method, while the multimodal extractor builds on existing RoBERTa and BiLSTM models, we have carefully designed its structure to effectively handle both visual content and long documents.
> For the interactive generator, we introduce an adaptive memory mechanism to address the challenges of generating long target sequences. Additionally, we design a layout-oriented pretraining approach to enhance layout prediction performance. The effectiveness of both the adaptive memory mechanism and the pretraining approach is validated through our ablation study.
>
>
> Q3: Only compare with one baseline.
>
> R3:  As a novel task, there are few feasible approaches that can be directly adapted to our scenario. Previous research on scientific poster generation has primarily focused on either poster composition or content extraction. The former proposed a simple probabilistic graphical model to infer panel attributes, but it requires human annotation for the panels of a poster. In contrast, our new dataset does not include panel information.
> For content extraction, we compare our model against two additional baselines: NeuralExt [1] and MSMO [2]. NeuralExt is a neural extractive model designed to extract text, figures, and tables from a paper. MSMO is a multimodal attention model that jointly generates text and selects the most relevant image from multimodal input. We adapted MSMO for the content extraction task. The experimental results are presented in the following table.
> |  Methods   | ROUGE-1 | ROUGE-2 | ROUGE-L|  ImgP | ImgR  |
> |  ----           | ----          | ----           | ----          | ----   | ----     |
> | NeuralExt   |     36.55   | 12.67       |  14.43      | 31.68 | 24.91 |
> | MSMO        |     32.45   | 10.43       |  12.51      | 36.64 | 32.56 |
> | AdaD2P      |     38.28   | 13.04       |  15.72      | 38.24 | 33.76 |
> | ours           |     40.68   | 14.76.        | 17.54        | 44.43 | 40.57 |
>
>
> Our model outperforms the baselines due to its hierarchical structure, which is better suited for handling long documents. In contrast, NeuralExt achieves the worst performance in image extraction, as it is a single-modal model that relies solely on figure and table captions to represent visual elements. We have updated the experimental result of multimodal content extraction in our uploaded revised version of paper.
> [1] Neural Content Extraction for Poster Generation of Scientific Papers
> [2] MSMO: Multimodal Summarization with Multimodal Output
>
> Q4: The generated posters are still poor and not suitable for practical applications.
>
> R4: We acknowledge that the proposed approach is still a step away from fully automated poster generation. Even state-of-the-art AI models, such as ChatGPT, can make errors when performing specific tasks. However, our method focuses on automatically generating editable draft posters from research papers, enabling researchers to make only minor adjustments to produce the final poster, thereby saving significant time.
> Scientific poster generation is inherently challenging, with several complexities, including diverse and intricate layouts, a large number of elements, multimodal processing, and handling lengthy inputs. To address these challenges, we aim to release a new dataset and introduce a baseline approach to advance research in the field of scientific poster generation.

---

> > ### Author Response · Authors · 2024-11-22
> >
> > Q5: Is it possible to incorporate prompts to generate the posters, allowing for personalized control by users over the final output?
> >
> > R5: Thanks for your suggestions. Before beginning our work, we also considered using a prompt to drive a large language model (LLM) for poster generation. However, this approach struggled with handling multimodal elements, such as scientific figures and tables, as well as generating flexible layouts. For layout generation, it might be feasible to combine predefined templates with the design capabilities of LLMs. Leveraging LLMs for fully automated poster generation is indeed another promising research direction, and we plan to explore it in future work.

---

> > > ### Author Response · Authors · 2024-11-29
> > >
> > > Dear reviewer,
> > >
> > > We sincerely appreciate your time and effort in reviewing our manuscript and offering valuable suggestions. As the author-reviewer discussion phase is drawing to a close, we would like to confirm whether our responses have effectively addressed your concerns. If you require further clarification, please do not hesitate to contact us.
> > >
> > > Best regards.

---

> > > > ### Comment · Reviewer_g553 · 2024-11-29
> > > >
> > > > Thanks for the responses. I have reviewed these responses and the comments from other reviewers. I have no further questions and would like to maintain my current positive rating.

---

### Official Review · Reviewer_Hgmt · 2024-11-04

**Soundness:** 3
**Presentation:** 3
**Contribution:** 3
**Rating:** 6
**Confidence:** 4

**Summary:**

This paper is focused on the task of scientific poster generation, and introduces two novel contributions in the field. First, a dataset of 10k examples with pairs of papers and posters (SciPG), with relevant annotations about content and layout. Second, they propose a new method for poster generation, designed for jointly generate the layout and the content of a poster. The method first extracts relevant texts and images from an input document (paper), by computing CLIP and RoBERTa embeddings, for images and texts respectively, and predicting an extractive score. They keep top-k from the predicted scores. Second, they leverage a BART model to process multimodal inputs and generate the summarized (paraphrased) texts and layout of the poster.

**Strengths:**

- The paper is generally well written and easy to follow.
- It develops new dataset with strong level of processing and curation. It also provides a good level of documentation on how to reproduce the processing pipeline. This dataset will be of great benefit to the community.
- The proposed method seems effective in addressing the limitations of previous works i.e. they focus on independently generating layout or content, and the proposed method performs both tasks.
- The paper has substantial experiments and ablations of the method.

**Weaknesses:**

- Authors claim this is a large-scale dataset, which might be not the best term to use, given the dimension of other datasets considered large scale (in the millions). A medium size dataset is more well suited.
- The paper does not argue about PosterLayout [1]. Even though it does not focus on scientific poser generation, it is good to have it as reference.
- There is just one baseline to compare. I understand this is a new task but authors should be absolutely certain that there are no other possible baselines. (See my question about this later)
- The qualitative samples in Figure 4 show that the texts are overlapping with each other, showing that the method has quite room for improvement.
- The human evaluation is done over 3 humans, which seems a rather small portion in order to draw conclusions.
-
[1] Hsu, Hsiao Yuan, et al. "Posterlayout: A new benchmark and approach for content-aware visual-textual presentation layout." Proceedings of the IEEE/CVF Conference on Computer Vision and Pattern Recognition. 2023.

**Questions:**

- About baselines. For instance, In Section 2.3 the paper describes that previous methods focus on layout or composition. Would it be possible to select a random layout and allow the baseline to generate its content? The same things applies the other way around, evaluating how the other baseline generates the layout. It would be interesting to compare both tasks separately.

- Can you elaborate more on why the KL term is needed in the generative loss? Authors mention " to prevent the model from
becoming overconfident." Some more detail would be appreciated.

- Authors define an Adaptive Memory mechanism to manage long range dependencies. Is this done because of conext length limitations in the Bert encoder? In that case, could this adaptive memory be avoided with sufficient context length?

(see other points commented in Weaknesses)

Format Questions
- I noticed you should use more the \citep{} command, when citing several works. They will appear better in the paper.
- Did the authors change some of the margins of the template? Some parts are quite to packed with content, seems too much use of vspace.

---

> ### Author Response · Authors · 2024-11-22
>
> Q1: The large-scale dataset problem.
>
> R1: Thank you for your suggestions. We have constructed a new dataset for scientific poster generation, consisting of over 10,000 (paper, poster) pairs, which is significantly larger than other existing datasets in this field. We have removed the qualifier 'Large-scale' in our revised version.
>
> Q2: The other work: PosterLayout
>
> R2: Thank you for your suggestions. PosterLayout focuses on advertisement poster design for products. Before presenting our work, we carefully examined the existing layout generation task, to which PosterLayout also belongs. This task typically involves predicting the position and category of graphic elements on an empty canvas, an image (e.g., advertising posters), or an image with text constraints. However, existing layout generation methods primarily focus on the element categories, generating layouts with different categories of elements without considering the actual textual or visual content within each element. Essentially, these methods create layout templates with different categories of bounding boxes, which can result in a misalignment between the predicted bounding boxes and the content inside them. For example, consider a case with 20 elements, including 15 text elements and 5 figures. While existing methods might generate 20 bounding boxes with visually coherent layouts based on element categories (e.g., text and figures), they fail to address which specific text or figure should be placed in which bounding box.
> In contrast, in our scientific posters, each element contains not only category information but also specific semantic content, which is crucial for ensuring the semantic consistency and coherence of the poster. Therefore, existing layout generation methods are insufficient for generating a feasible scientific poster.
>
> We have clarified the difference between our work and the existing layout generation task in our related work.
>
>
> Q3: Would it be possible to select a random layout and allow the baseline to generate its content? The same things applies the other way around, evaluating how the other baseline generates the layout.
>
> R3:Thank you for your suggestions. In our scientific poster generation task, the layout of each element depends on its corresponding content. Adopting a completely isolated layout prediction approach is infeasible for generating a semantically coherent poster. For instance, a layout prediction model might generate a bounding box that is too small to accommodate a long sentence if it ignores the semantics of the given sentence.
> What you mentioned aligns with dividing our task into three subtasks: content extraction, content paraphrasing, and layout prediction. In fact, our baseline model, AdaD2P, follows such a three-phase approach for poster generation. The layout of element is based on the current context and the content within the element. However, our proposed method unifies content paraphrasing and layout prediction into a single subtask to better capture both the semantic representation within each element and the relationships among elements.
> In the evaluation, layout and content are also assessed separately in our paper.
>
>
>
> Q4:  The texts are overlapping with each other in the qualitative samples
>
> R4: Thanks for your comments. We acknowledge that the proposed approach is still a step away from fully automated poster generation. Even state-of-the-art AI models, such as ChatGPT, can make errors when performing specific tasks. However, our method focuses on automatically generating editable draft posters from research papers, enabling researchers to make only minor adjustments to produce the final poster, thereby saving significant time.
> Scientific poster generation is inherently challenging, with several complexities, including diverse and intricate layouts, a large number of elements, multimodal processing, and handling lengthy inputs. To address these challenges, we aim to release a new dataset and introduce a baseline approach to advance research in the field of scientific poster generation.
>
>
> Q5: Human evaluation.
>
> R5: Thanks for your suggestions. In addition to the three existing human raters, we also invited seven additional raters to conduct the human evaluation. The results are updated in Figure 3 in our revised version of our paper.  From Figure 3, we observe that the new evaluation results are on par with the previous ones.
>
> Q6: The KL term in the generative loss.
>
> R6: Sorry for the confusions. We include the KL-Divergence term in our training objectives because we observed that the training objective often suffers from overfitting, particularly in the bounding box predictions for layouts. To address this issue, we minimize the KL-Divergence between the softmax predictions (prediction distributions) and the one-hot output distribution (ground truth) with Label Smoothing. This approach helps mitigate the overfitting problem. We have clarified this in our revised version.

---

> > ### Author Response · Authors · 2024-11-22
> >
> > Q7: The context length limitations
> >
> > R7: Sorry for your confusions. The standard BERT model can process a maximum of 512 tokens. However, both our input and target token lengths exceed 1,000 tokens. To handle these long sequences, we splits them into segments and passes memory states from the previous segment to the current one. The proposed adaptive memory passing mechanism makes the model recurrent, effectively removing the input sequence length limitation. While the RMT model can theoretically handle sequences of infinite length, in practice, it is constrained by memory capacity and the efficiency of memory access and update operations. We have clarified this in the revised version of our paper.
> >
> >
> > Q8: Format Questions
> >
> > R8: Thank you for pointing out the formatting issues in our paper. We have thoroughly reviewed and addressed these concerns, and have uploaded a revised version of the paper to the submission system.

---

> > > ### Comment · Reviewer_Hgmt · 2024-11-25
> > >
> > > Thanks to the authors for their responses. My concerns have been clarified, and this appears to be solid work in the field of poster generation. A few recommendations:
> > >
> > > 1. The paper still feels too packed and would benefit from some additional space.
> > > 2. I suggest including more qualitative results comparing the method to the baseline, along with a discussion of common failure cases or the strengths of the methods.
> > >
> > > That said, I am happy to maintain my score.

---

> > > > ### Author Response · Authors · 2024-11-25
> > > >
> > > > Thank you for your suggestions. We will re-optimize the layout of our paper. Additionally, we will include more qualitative results and examples of common failure cases in the revised manuscript.

---

### Note · Authors · 2025-01-23

I have read and agree with the venue's withdrawal policy on behalf of myself and my co-authors.